# Impaired estimated glomerular filtration rate and associated factors among adult patients living with HIV at Asella Referral and Teaching Hospital, Ethiopia: A cross-sectional study

**Tamiru Adugna Dadi**[1]*, **Teshome Tola Bedada**[1]*, **Abdurke Dido Akako**[1],
**Legesse Tadesse Wodajo**[2], **Sebona Girma Moges**[1], **Wubshet Abraham Alemu**[1]

**1** Department of Internal Medicine, College of Health Sciences, Arsi University, Asella, Oromia, Ethiopia,
**2** Department of Public Health, College of Health Sciences, Arsi University, Asella, Oromia, Ethiopia

* adutamiru@gmail.com (TAD); tolateshome@gmail.com (TTB)

## Abstract

### Background

Human Immunodeficiency Virus (HIV) remains a major global health challenge, particularly in sub-Saharan Africa. It is a multisystem disorder with significant renal involvement. In Ethiopia, limited data exist on the prevalence of impaired estimated glomerular filtration rate among patients with Human Immunodeficiency Virus. Identifying associated factors is critical for implementing targeted interventions to preserve renal function.

### Methods

An institution-based cross-sectional study was conducted among 252 adult patients living with Human Immunodeficiency Virus from October 1 to December 30, 2024 at Asella Referral and Teaching Hospital. Data were collected via structured questionnaires and medical record reviews. Bivariate and multivariate logistic regression analyses were performed to identify factors associated with impaired glomerular filtration rate (estimated glomerular filtration rate <60 mL/min/1.73m$^2$).

### Result

The prevalence of impaired glomerular filtration rate was 18.7% (95% CI: 14.0–23.0). Factors significantly associated with impaired glomerular filtration rate included age > 40 years (AOR = 3.26; 95%CI: 1.17–9.12), history of smoking (AOR = 4.68; 95% CI: 1.87–11.70), opportunistic infections (AOR = 5.93; 95% CI: 2.23–15.74), diabetes mellitus (AOR = 3.86; 95% CI: 1.47–10.12) and hypertension (AOR = 2.71; CI: 1.07–6.82).

**Data availability statement:** All relevant data are within the manuscript and its Supporting Information files.

**Funding:** The author(s) received no specific funding for this work.

**Competing interests:** The authors have declared that no competing interests exist.

**Abbreviations:** AIDS, Acquired Immunodeficiency Syndrome; AKI, Acute Kidney Injury; AOR, Adjusted Odd Ratio; ART, Anti-retroviral therapy; ATRH, Asella Teaching and referral hospital; BMI, Body Mass Index; BP, Blood Pressure; CD, Cluster of differentiation; CG, Cockcroft-Gault; CI, Confidence Interval; CKD, Chronic Kidney Disease; CKD-EPI, Chronic Kidney Disease Epidemiology; COR, Crude Odd Ratio; DM, Diabetes Mellitus; eGFR, estimated glomerular filtration rate; eGFRcr, Creatinine based estimated glomerular filtration rate; eGFRcys, Cystatin based estimated glomerular filtration rate; ERC, Ethical Review Committee; ESRD, End Stage Renal Disease; ETB, Ethiopian Birr; HDU, High Dependency Unit; HIV, Human Immunodeficiency Virus; HIVAN, HIV associated nephropathy; ICU, Intensive Care Unit; MDRD, Modification of Diet in Renal Disease Study; OR, Odds Ratio; ORCID, Open Researcher and Contributor ID; OPD, Outpatient Department; PLHIV, People Living with HIV; Ref, Reference; SD, Standard Deviation; SPSS, Statistical Package for the Social Sciences; TB, Tuberculosis; WHO, World Health Organization.

## Conclusion

The study found that around 19% of the studied population had impaired glomerular filtration rate. Targeted screening for kidney disease should focus on individuals older than 40years, smokers and those with opportunistic infections, diabetes mellitus or hypertension.

---

## Background

Human Immunodeficiency Virus (HIV) remains a major global health challenge, particularly in sub-Saharan Africa, where it continues to be a leading cause of morbidity and mortality [1–5]. HIV is a multisystem disorder with significant renal involvement. People living with HIV (PLHIV) demonstrate diverse renal pathology, including acute kidney injury (AKI), HIV-associated nephropathy (HIVAN), chronic kidney disease (CKD), and antiretroviral therapy (ART)-induced nephrotoxicity. Although therapeutic advances have improved outcomes, CKD and end-stage renal disease (ESRD) remain disproportionately prevalent in this population. The widespread use of ART has fundamentally altered the spectrum of HIV-related kidney disease, marked by declining incidence of classic HIVAN but increasing burden of age-related comorbidities and ART-associated nephrotoxicity [6–8].

Glomerular filtration rate (GFR) is the most useful measurement of kidney function. It is essential for diagnosing, staging, and managing chronic kidney disease (CKD). Current guidelines recommend estimated GFR (eGFR) over serum creatinine or cystatin C levels, favoring eGFR based on creatinine (eGFRcr) or cystatin C (eGFRcys) in most cases [9,10].

Based on creatinine-based estimated glomerular filtration rate, chronic kidney disease (CKD) is classified into five stages: stage 1 (kidney damage with normal or increased eGFR ≥ 90 mL/min/1.73 m$^2$), stage 2 (mild reduction in eGFR 60–89), stage 3 (moderate reduction, subdivided into 3a [eGFR 45–59] and 3b [30–44]), stage 4 (severe reduction, eGFR 15–29), and stage 5 (kidney failure, eGFR < 15 or dialysis dependence). This staging system reflects progressive loss of renal function and guides clinical management, with stages 3–5 indicating progressively greater impairment that typically requires therapeutic intervention and monitoring for complications [10,11].

The Chronic Kidney Disease Epidemiology Collaboration (CKD-EPI) equation, which incorporates serum creatinine level and demographic factors, appears to provide the most accurate estimates among HIV-infected persons who are stable on antiretroviral therapy [8,10–12].

Among adults with HIV, chronic kidney disease (CKD) prevalence shows significant variation depending on estimation method and geographic region. When assessed using different equations, prevalence rates are: 6.4% (MDRD), 4.8% (CKD-EPI), and 12.3% (Cockcroft-Gault). Regional disparities are evident, with North America (6.5%), South America (6.2%), and Europe (2.7%) showing relatively lower rates compared to sub-Saharan Africa, where prevalence ranges dramatically from 25% to 77% [4,13].

Among people living with HIV in Ethiopia, the reported prevalence of renal impairment varies substantially across studies, ranging from 4% to 30% [14,15].

Early detection of chronic kidney disease (CKD) through screening for reduced estimated glomerular filtration rate (eGFR) and/or proteinuria enables timely clinical intervention, potentially mitigating the associated risks of cardiovascular events, kidney failure, and mortality [4,10,16,17].

Undiagnosed renal impairment significantly worsens clinical outcomes, particularly in resource-limited settings where access to renal replacement therapy remains constrained. This study addresses critical evidence gaps regarding estimated glomerular filtration rate (eGFR) impairment and associated factors among people living with HIV in Ethiopia. The findings will provide actionable insights to guide clinical management, health policy formulation, and targeted research aimed at improving renal health outcomes in this vulnerable population.

## Method and materials

### Study area and period

This study was conducted at Asella Referral and Teaching Hospital (ARTH) in the Arsi Zone of Oromia, approximately 175 km southeast of Addis Ababa, Ethiopia. The study was conducted from October 01, 2024 to December 30, 2024.

Established as one of the region's earliest public hospitals, ARTH is centrally located in Asella town on a 4.3-hectare campus. It delivers preventive, curative, and diagnostic services to residents in Asella and surrounding areas while functioning as a major referral center.

The hospital offers a wide range of specialized services through its emergency department, outpatient (OPD) unit, and dedicated departments for internal medicine, pediatrics, gynecology & obstetrics, surgery, oncology, dermatology, radiology, ophthalmology, psychiatry, and critical care (ICU). It also includes a specialized fistula unit. The internal medicine department is organized into: female ward (16 general beds, 6 TB beds, and 2 private/isolation beds), male ward (18 general beds, 6 TB beds, and 2 private/isolation beds), and critical care units (A 4-bed high-dependency unit (HDU) and a 6-bed ICU).

The department gives specialized follow-up clinics for chronic conditions, including diabetes, neurological disorders, cardiovascular diseases, and infectious diseases (such as HIV/AIDS) by a multidisciplinary team of cardiologists, neurologists, internists, resident physicians and general practioners.

As a public teaching hospital, ARTH provides comprehensive HIV/AIDS care, with 3,886 active ART patients (3,749 adults) during the study period. According to the Central Statistical Agency (2007), the hospital serves nearly seven million people across the Arsi Zone.

### Study design

An institution based cross-sectional study design was employed.

### Source population

All adults living with HIV (age ≥ 18 years) receiving ART care at the ARTH clinic were source population.

### Study population

The study enrolled adults living with HIV, aged 18–65 years, who were on follow-up at the ARTH ART clinic during the study period. Individuals over 65 years were excluded to minimize potential confounding from age-related declines in glomerular filtration rate (GFR), which are distinct from HIV-associated kidney dysfunction.

### Inclusion criteria

All adults living with HIV who attended routine follow-up visits at the ART clinic during the study period were included.

## Exclusion Criteria

We excluded adults living with HIV who were: (1) acutely ill or medically unstable, (2) unable to provide informed consent, (3) pregnant during the study period, or (4) had known pre-existing kidney disease.

## Sample size determination

Sample size was determined using single population formula and considering the following assumptions: 95% confidence interval, a margin of error of 5%, and a 20.7% (P) prevalence of impaired eGFR in Ethiopia based on study from Mettu Karl Referral Hospital [18]. We used the correctional formula since our source population is less than 10,000. Considering 10% nonresponse rate, the final sample size of was calculated to be 260.

## Sampling procedure

Participants were selected via systematic random sampling. From a population of 3,749, a sampling interval (K) of 14 (K = N/n) was calculated, and a random start point between 1 and 14 was chosen. Every 14th patient was subsequently enrolled until the target sample size (n = 260) was reached.

## Data collection tool

Data collection tool (S1 in S1 File) was adapted from similar study conducted in Mettu Kerl hospital [18]. It was translated to local languages and back-translated to English to check for the consistency. Data on socio-demographic and behavioral factors likes smoking, alcohol consumption, diabetic mellitus, hypertension, age, sex, occupation and marital status was collected via interview using structured questionnaire.

Relevant clinical data, including CD4 cell count, serum creatinine levels, WHO clinical staging, and ART regimen, were extracted from patient medical records. Serum creatinine laboratory measurements were performed using a linear enzymatic reagent method. Renal function was assessed by estimating glomerular filtration rate (eGFR) using the Chronic Kidney Disease Epidemiology Collaboration 2021 (CKD-EPI 2021) equation to the determined serum creatinine values.

## Data quality control

Prior to data collection, a one-day training session was conducted for three nurse data collectors and one supervisor (a second-year medical resident). The training covered the proper use of data collection tools, standardized interpretation of questionnaire items, clear communication of study objectives, implementation of sampling techniques, and maintenance of participant confidentiality.

Data collectors were also trained on the study's scientific and clinical relevance, ethical considerations including participant rights and confidentiality and best practices for conducting structured interviews.

A designated supervisor provided continuous oversight, and collected data were reviewed daily for completeness and internal consistency.

## Data entry and analysis

The collected data were entered into Epi data version 4.6 and analyzed via Statistical Package for the Social Sciences (SPSS) version 26.0 statistical packages. Descriptive statistical analysis was conducted to describe the number and percentage of the variables. Binary analysis was performed to examine the unadjusted relationships between the predictors and dependent variables. A bivariate logistic regression model was used to identify factors associated with the outcome variable. Those variables with a P value of < 0.25 were subsequently entered into multiple logistic regressions to identify independent predictors. The model goodness of fit was tested by the Hosmer and Lemeshow technique, and the presence of multicollinearity was checked by estimating variance inflation factors. The strength of the association was assessed

using adjusted odds ratios (AOR) with 95% confidence intervals (CI), and statistical significance was determined at a p-value of < 0.05.

## Study variables

**Dependent variable.** Estimated glomerular filtration.

**Independent variables. Socio-demographic variables**: age, sex, residence, educational level, occupation, marital status, income, family history of renal disease.

**Behavioral factors**: cigarette smoking, alcohol, consumption.

**Medical (HIV-related) variables**: opportunistic infections, comorbidity (diabetes, hypertension), Tenofivir-based regimen, CD4 counts, body mass index, WHO stage.

## Operational definition

**Impaired glomerular filtration rate:** defined as a significant reduction in creatinine-based estimated glomerular filtration rate (eGFR) < 60 ml/min/1.73m2 [10,11].

**Adult**: An individual aged 18 years or older.

**Current smoking**: at least one cigarette per day in the past 30 day.

**Comorbidity**: an individual with two or more coexisting chronic non-communicable diseases (NCDs) confirmed by medical diagnosis or standardized criteria.

## Ethical consideration

Ethical issues were addressed at all stages of the research process. Ethical clearance was obtained from Ethical Review Committee (ERC) of Arsi University College of Health Sciences (A/CHS/RC/131/2024). Detailed information about the purpose of the study was provided to participants in their local language. They were informed about why they were selected and were assured that their participation were entirely voluntary. The study did not involve any unfamiliar procedures and had no risks to the participants.

The participants were guaranteed that their access to healthcare services remained the same, regardless of their decision to participate. They were also notified that data collectors would extract relevant information from their medical records.

All the collected data were used for research purposes. Confidentiality and anonymity were maintained by recording only necessary information and excluding personal identifiers. Finally, written informed consent was obtained for those who can read and write while oral consent obtained and signed by witnesses for illiterate participants prior to data collection.

## Result

### Socio demographic characteristics

A total of 252 individuals participated in the study, yielding a response rate of 96.9%. More than half of the participants (54.8%) were female, with a mean age of 44 years (±10.8 SD). The majority were married (52%) and self-employed (48%). Seventy-five percent of respondents lived in urban areas, and most were followers of the Orthodox religion (47.2%). In terms of education, 29.4% had attained primary education, while 52% of participants earned more than 3,500 ETB per month (Table 1).

### Anthropometric measurements

BMI was derived from standardized height and weight measurements obtained with participants barefoot. According to the analysis, 64.7% of participants fell within the normal weight range, whereas 22.6% were classified as underweight, suggesting nutritional deficiency.

**Table 1. Socio-demographic characteristics of study participants, Asella Referral and Teaching Hospital, Ethiopia (N = 252).**

| Variable | Category | Frequency | Percentage (%) |
|---|---|---|---|
| **Sex** | Male | 114 | 45.2 |
| | Female | 138 | 54.8 |
| **Age (years)** | 18–39 | 84 | 33.3 |
| | 40–65 | 168 | 66.7 |
| **Marital status** | Single | 48 | 19.0 |
| | Married | 131 | 52.0 |
| | Divorced | 39 | 15.5 |
| | Widowed | 34 | 13.5 |
| **Educational status** | No formal education | 32 | 12.8 |
| | Primary education | 74 | 29.4 |
| | Secondary education | 73 | 29.0 |
| | College & above | 73 | 29.0 |
| **Religion** | Orthodox | 119 | 47.2 |
| | Muslim | 92 | 36.5 |
| | Protestant | 41 | 16.3 |
| **Occupation** | Self-employed | 121 | 48.0 |
| | Government employee | 50 | 19.8 |
| | Unemployed | 52 | 20.6 |
| | Student | 12 | 4.8 |
| | Daily laborer | 17 | 6.7 |
| **Residency** | Urban | 190 | 75.4 |
| | Rural | 62 | 24.6 |
| **Monthly income (ETB)** | <3,500 | 121 | 48.0 |
| | ≥3,500 | 131 | 52.0 |

### Behavioral characteristics

Participants were interviewed regarding their cigarette smoking and alcohol consumption behaviors. The results showed that 15.9% reported a history of smoking, with 6.0% being active smokers. Additionally, nearly 22% of respondents consumed alcohol, of whom 9.9% drank once or twice weekly (Table 2).

### Immunological and clinical characteristics

The mean CD4 count among participants was 415.2 ± 184.9 cells/mm$^3$. The majority (91.3%) had CD4 counts >200 cells/mm$^3$, while 8.7% showed advanced immunosuppression (<200 cells/mm$^3$). Regarding WHO clinical staging, 43.3% were classified as stage I (Fig 1). Functional status assessment revealed that 91.7% (n = 231) maintained working capacity, while 8.3% (n = 21) were ambulatory.

### Antiretroviral therapy characteristics

The majority of participants (88.1%) were receiving first-line ART regimens, with 91.7% adhering to a once-daily dosing schedule. Among those on ART, 71.4% were prescribed tenofovir-based regimens. Nearly four-fifths of participants (77.8%, n = 196) reported no history of treatment interruption (Table 3).

**Table 2. Substance use characteristics among study participants, Asella Referral and Teaching Hospital, Ethiopia (N = 252).**

| Variable | Frequency | Percentage (%) |
|---|---|---|
| **Tobacco use history** | | |
| Never used | 212 | 84.1 |
| Ever used | 40 | 15.9 |
| **Frequency among ever users** | | |
| Monthly | 4 | 1.6 |
| Weekly | 5 | 2.0 |
| 1-2 times weekly | 31 | 12.3 |
| **Current tobacco use** | | |
| No current use | 237 | 94.0 |
| Current use | 15 | 6.0 |
| **Frequency among current users** | | |
| Monthly | 4 | 1.6 |
| 1-2 times weekly | 11 | 4.4 |
| **Alcohol consumption** | | |
| Never consumes | 197 | 78.2 |
| Ever consumes | 55 | 21.8 |
| **Frequency among drinkers** | | |
| Weekly | 6 | 2.4 |
| 1-2 times weekly | 25 | 9.9 |
| Occasionally | 24 | 9.5 |

## Comorbidities and opportunistic infections

The study found that around three percent (2.8%) of participants reported a family history of renal disease. Approximately one-third (33.3%) had experienced opportunistic infections, while comorbid conditions included diabetes mellitus (15.9%) and hypertension (18.3%) (Table 4).

## Estimated glomerular filtration rate

The mean estimated glomerular filtration rate (eGFR) was $90.5 \pm 30.3$ mL/min/1.73m$^2$. Analysis revealed that more than half (53.6%) of study participants maintained preserved renal function (eGFR > 90 mL/min/1.73m$^2$), whereas 18.7% (95% CI: 14–23%) exhibited clinically significant renal impairment. A small proportion (0.8%, n = 2) met criteria for kidney failure (eGFR < 15 mL/min/1.73m$^2$) (Fig 2).

## Factors associated with impaired eGFR

In our bivariate analysis examining predictors of impaired eGFR, we evaluated twenty clinical and demographic variables for potential associations. These included demographic factors (age, sex, residence, education level, occupation, marital status, income), clinical characteristics (family history of renal disease, BMI, WHO clinical stage, CD4 count), comorbidities (diabetes, hypertension, opportunistic infections), behavioral factors (smoking history, alcohol use), and ART-related variables (tenofovir-based regimen, regimen category, dosing frequency, treatment interruption history). Eleven variables showing preliminary associations (P < 0.25) were subsequently included in our multivariable logistic regression model for further analysis.

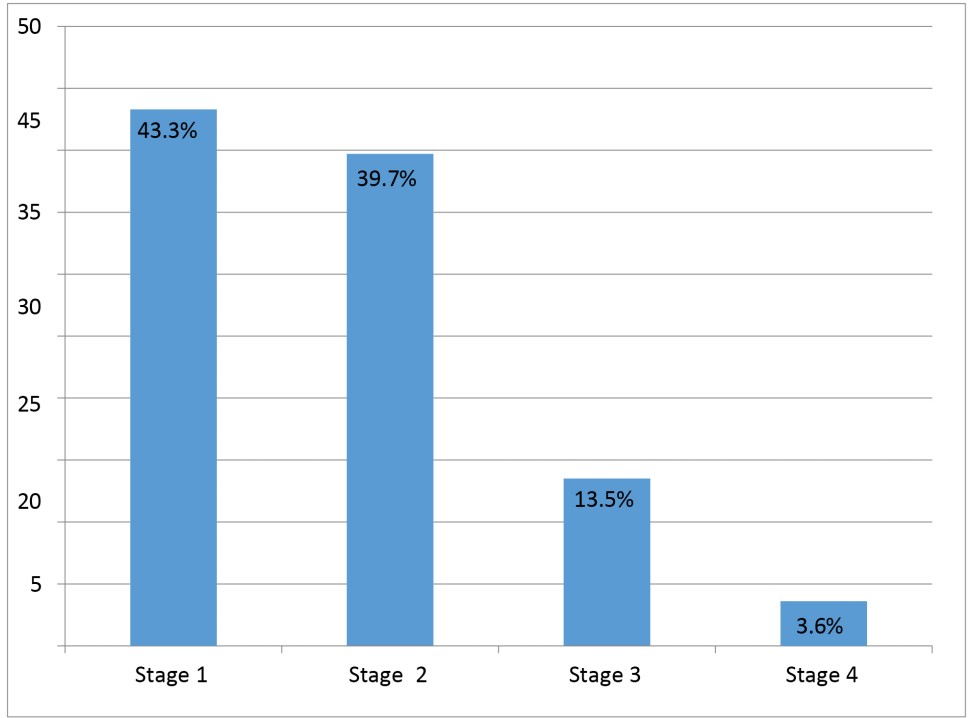

**Fig 1. WHO Clinical stages of adult patients living with HIV, Asella Referral and Teaching Hospital, Ethiopia (N = 252).**

**Table 3. Antiretroviral therapy characteristics of study participants, Asella Referral and Teaching Hospital, Ethiopia (N = 252).**

| Variable | Category | Frequency | Percentage (%) |
|---|---|---|---|
| **ART regimen line** | First-line | 222 | 88.1 |
| | Second-line | 27 | 10.7 |
| | Third-line | 3 | 1.2 |
| **Dosing frequency** | Once daily | 231 | 91.7 |
| | Twice daily | 18 | 7.1 |
| | Three times daily | 3 | 1.2 |
| **Tenofovir-based regimen** | No | 72 | 28.6 |
| | Yes | 180 | 71.4 |
| **Treatment interruptions** | Experienced | 56 | 22.2 |
| | None reported | 196 | 77.8 |

After adjustment for potential confounders, five factors emerged as significant independent predictors of impaired estimated glomerular filtration rate (eGFR). Participants aged 40 years and above had 3.3-fold higher odds of impaired GFR compared to younger individuals (AOR = 3.26; 95%CI: 1.17–9.12). Current or former smokers demonstrated substantially increased risk, with 4.7 times greater odds of impaired GFR than non-smokers (AOR = 4.68; 95%CI: 1.87–11.70). The presence of opportunistic infections showed the strongest association, conferring a 5.9-fold elevated risk (AOR = 5.93; 95%CI: 2.23–15.74). Additionally, diabetes mellitus (AOR = 3.86; 95%CI: 1.47–10.12) and hypertension (AOR = 2.71; 95%CI: 1.07–6.82) were independently associated with 3.9-fold and 2.7-fold increased odds of impaired GFR, respectively (Table 5).

**Table 4. Comorbidities and opportunistic infections among study participants, Asella Referral and Teaching Hospital, Ethiopia (N = 252).**

| Variable | Category | Frequency | Percentage (%) |
|---|---|---|---|
| **Familial renal disease** | Yes | 7 | 2.8 |
| | No | 245 | 97.2 |
| **Opportunistic infections** | Yes | 74 | 29.4 |
| | No | 178 | 70.6 |
| **Diabetes mellitus** | Yes | 40 | 15.9 |
| | No | 212 | 84.1 |
| **Hypertension** | Yes | 46 | 18.3 |
| | No | 206 | 81.7 |

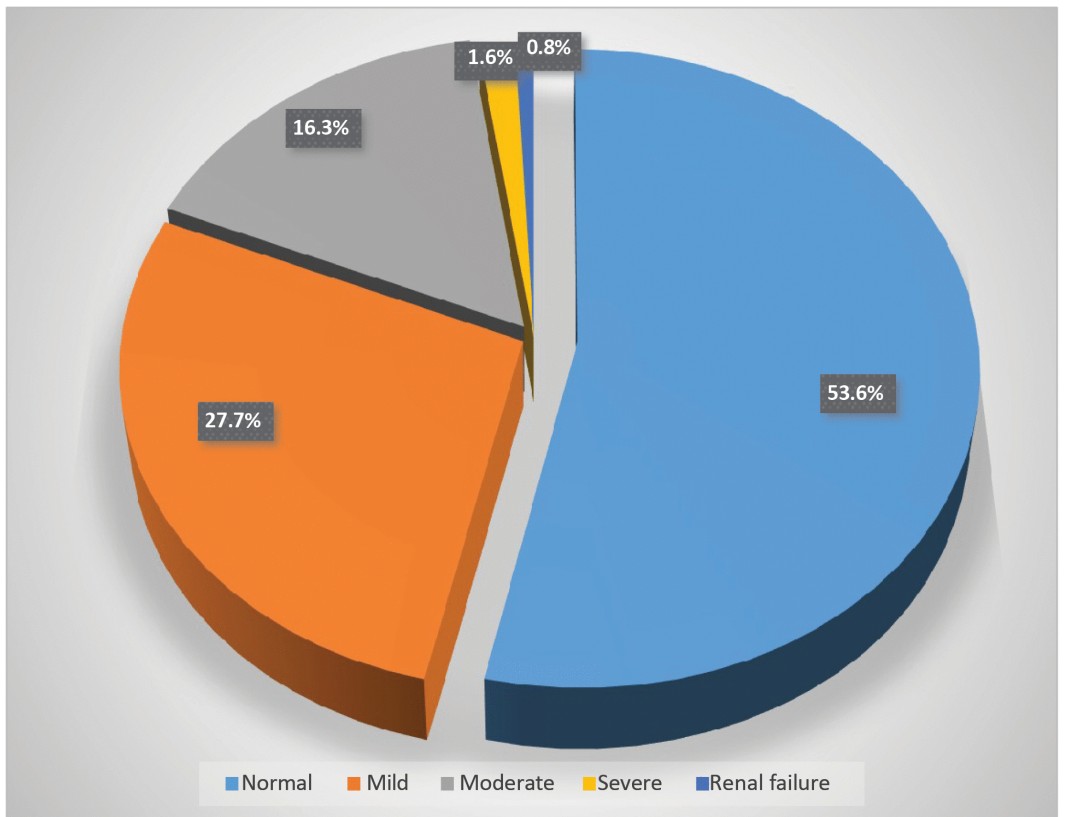

**Fig 2. Estimated glomerular filtration rate of adult patients living with HIV, Asella Referral and Teaching Hospital, Ethiopia (N = 252).**

## Discussion

This study assessed the magnitude of impaired estimated glomerular filtration rate (eGFR) and its associated factors among adult HIV patients. The findings revealed that 18.7% (95% CI: 14–23) of study participants had impaired eGFR, highlighting a significant burden of renal dysfunction in this population. This finding aligns closely with rates reported in similar Ethiopian settings, including studies from Mettu Karl Referral Hospital [18] and Gondar Referral Hospital [19], reinforcing the consistency of this health challenge across different regions.

**Table 5. Factors associated with impaired estimated glomerular filtration rate among HIV positive adults at Asella Referral and Teaching Hospital, Ethiopia (N = 252).**

| Variables | Impaired GFR n = 47 (%) | Normal GFR n = 205 (%) | Crude OR (95% CI) | P-value | Adjusted OR (95% CI) | P-value |
|---|---|---|---|---|---|---|
| **History of smoking** | | | | | | |
| Yes | 16 (34.0) | 24 (11.7) | 3.89 (1.86–8.14) | <0.001 | 4.68 (1.87–11.70) | 0.001* |
| No | 31 (66.0) | 181 (88.3) | 1 [Ref] | — | 1 [Ref] | — |
| **Opportunistic infections** | | | | | | |
| Yes | 26 (55.3) | 48 (23.4) | 4.05 (2.07–8.03) | <0.001 | 5.93 (2.23–15.74) | <0.001* |
| No | 21 (44.7) | 157 (76.6) | 1 [Ref] | — | 1 [Ref] | — |
| **Diabetes mellitus** | | | | | | |
| Yes | 14 (29.8) | 26 (12.7) | 2.92 (1.38–6.17) | 0.005 | 3.86 (1.47–10.12) | 0.006* |
| No | 33 (70.2) | 179 (87.3) | 1 [Ref] | — | 1 [Ref] | — |
| **Hypertension** | | | | | | |
| Yes | 14 (29.8) | 32 (15.6) | 2.29 (1.10–4.76) | 0.026 | 2.71 (1.07–6.82) | 0.034* |
| No | 33 (70.2) | 173 (84.4) | 1 [Ref] | — | 1 [Ref] | — |
| **Age (years)** | | | | | | |
| 18–39 | 8 (17.0) | 76 (37.1) | 1 [Ref] | — | 1 [Ref] | — |
| 40–65 | 39 (83.0) | 129 (62.9) | 2.87 (1.27–6.46) | 0.011 | 3.26 (1.17–9.12) | 0.024* |
| **CD4 count (cells/mm³)** | | | | | | |
| <200 | 10 (21.3) | 12 (5.9) | 4.34 (1.75–10.79) | 0.002 | 3.49 (0.71–17.14) | 0.124 |
| ≥200 | 37 (78.7) | 193 (94.1) | 1 [Ref] | — | 1 [Ref] | — |
| **ART interruption** | | | | | | |
| Yes | 17 (36.2) | 39 (19.0) | 2.41 (1.21–4.80) | 0.012 | 0.65 (0.20–2.05) | 0.463 |
| No | 30 (63.8) | 166 (81.0) | 1 [Ref] | — | 1 [Ref] | — |
| **Tenofovir regimen** | | | | | | |
| Yes | 28 (59.6) | 152 (74.1) | 0.51 (0.26–0.99) | 0.048 | 0.61 (0.24–1.51) | 0.287 |
| No | 19 (40.4) | 53 (25.9) | 1 [Ref] | — | 1 [Ref] | — |
| **WHO stage** | | | | | | |
| Stage 1 | 18 (38.3) | 91 (44.4) | 1 [Ref] | — | 1 [Ref] | — |
| Stage 2 | 12 (25.5) | 88 (42.9) | 0.68 (0.31–1.51) | 0.35 | 0.73 (0.27–1.97) | 0.538 |
| Stage 3–4 | 17 (36.2) | 26 (12.7) | 3.30 (1.49–7.30) | 0.003 | 0.63 (0.15–2.55) | 0.519 |
| **ART regimen** | | | | | | |
| First line | 37 (78.7) | 185 (90.2) | 1 [Ref] | — | 1 [Ref] | — |
| Second/Third line | 10 (21.3) | 20 (9.8) | 0.40 (0.17–0.92) | 0.032 | 1.87 (0.30–11.58) | 0.499 |
| **ART dose frequency** | | | | | | |
| Once daily | 40 (85.1) | 191 (93.2) | 1 [Ref] | — | 1 [Ref] | — |
| ≥2 times daily | 7 (14.9) | 14 (6.8) | 2.38 (0.90–6.29) | 0.078 | 0.76 (0.10–5.40) | 0.787 |

OR: odds ratio; CI: confidence interval; ART: antiretroviral therapy; WHO: World Health Organization; *Significant at p < 0.05.

Our study found a higher proportion of impaired eGFR (18.7%) compared to reports from Tanzania (6%), Nigeria (13.8%), Japan (13–14%), and Mexico (11.7%) [20–23]. These disparities likely reflect differences in: (1) population demographics and socioeconomic contexts, as these studies represent diverse geographic settings; (2) diagnostic criteria for impaired eGFR determination; and (3) healthcare system factors including HIV treatment protocols, healthcare infrastructure, and comorbidity prevalence (particularly hypertension and diabetes). These cross- national variations emphasize the importance of developing context-specific guidelines for renal monitoring in HIV care programs, particularly in sub-Saharan African settings like Ethiopia where both HIV burden and renal disease risk factors may differ substantially from other regions.

Increasing age was significantly associated with impaired eGFR, with participants aged ≥40 years demonstrating 3.26-fold higher odds of renal dysfunction compared to younger individuals (AOR = 3.26; 95% CI: 1.17–9.12). Consistent with previous multinational research from Ethiopia, Uganda, and Japan, these findings strengthen the evidence for an association between advanced age and renal impairment in people living with HIV [14,20,21,24,25].

This likely results from: natural age-related kidney decline, kidney damage from long-term HIV medications, more frequent health issues like diabetes and hypertension in older adults, and lasting inflammation from HIV that speeds up kidney aging.

The analysis revealed that participants with a history of smoking had 4.68 times higher odds of developing impaired eGFR compared to non-smokers (AOR = 4.68; 95% CI: 1.87–11.70). This significant association aligns with previous research conducted across diverse settings, including studies from Mettu Karl Referral Hospital in Ethiopia, and Aleppo University in Syria [18,26]. The later study was conducted among the general population. The consistent findings across these geographically distinct populations highlight smoking as a significant modifiable risk factor for renal dysfunction in this patient group. Smoking contributes to the deterioration of kidney function by increasing the risk of micro-albuminuria and accelerating its progression to high grade albuminuria, particularly in individuals with diabetes. Additionally, smoking induces inflammation, reduces renal blood flow, and promotes kidney aging, thereby elevating the risk of kidney disease.

Diabetes mellitus (DM) was significantly associated with impaired eGFR in our study population. Participants with DM had 3.86-fold higher odds of developing impaired eGFR compared to those without DM (adjusted OR = 3.86; 95% CI: 1.47–10.12). These findings align with evidences from clinical studies conducted in Ethiopia, Tanzania, and Japan, reinforcing the consistent relationship between DM and reduced renal function [18,20,22]. HIV-positive participants with diabetes were more likely to experience reduced GFR. Diabetes impairs glomerular filtration by damaging small blood vessels, overloading cells with excessive glucose, and triggering inflammation and scarring.

Hypertension was significantly associated with impaired glomerular filtration rate (GFR) among HIV-positive adults. Patients with hypertension had 2.71 times higher odds of GFR impairment compared to those without hypertension (AOR = 2.71; 95% CI: 1.07–6.82), underscoring how the coexistence of hypertension and HIV heightens the risk of renal dysfunction. This finding concurs with previous research conducted at Mettu Karl Hospital and in Japan [18,20]. Hypertension is a well-established risk factor for reduced GFR because persistently elevated blood pressure damages the glomeruli, progressively lowering GFR.

In our study, impaired glomerular filtration rate (GFR) showed a strong association with opportunistic infections: individuals with such infections had 5.93 times higher odds of impaired GFR (AOR = 5.93; 95% CI: 2.23–15.74). This finding is consistent with the cross-sectional study by Cailhol et al. in Burundi, which reported a high prevalence of chronic kidney disease largely driven by factors such as prior tuberculosis and urinary abnormalities in HIV-positive patients [27]. Opportunistic infections likely contribute to renal impairment through mechanisms including renal inflammation and nephrotoxic effects of antimicrobial treatments, which together exacerbate reductions in GFR.

Unlike prior studies, we did not observe a significant association between ART regimen and estimated GFR. As nephrotoxic effects are often cumulative, the duration of ART exposure which is a critical determinant of renal outcomes was not assessed in our study. This methodological limitation likely accounts for our null finding rather than indicating a true absence of effects.

This study has assessed comprehensive variables but is not without limitations: the cross-sectional design precludes causal inferences; recruitment from a single center limits generalizability; and renal function estimation was based on a single serum creatinine measurement, which may not reflect long-term kidney function. Moreover, residual confounding factors may exist despite multivariable adjustment. Therefore, future studies should adopt longitudinal designs, include diverse populations across multiple centers, and incorporate repeated measures of renal function over time to better evaluate causality and enhance external validity.

## Conclusion

Approximately 19% of the study population had impaired estimated glomerular filtration rate (eGFR). Targeted screening and early interventions for kidney disease should prioritize individuals living with HIV who are over 40 years of age, have a history of smoking, or present with opportunistic infections, diabetes mellitus, or hypertension.

## Supporting information

**S1 File. S1 Impaired eGFR Data Collection Tool. S2 Impaired eGFR among HIV source data file. S3 Table 5 Bivariate analysis of factors associated with impaired glomerular filtration rate. S4 Table 6 Multivariable analysis of factors associated with impaired estimated glomerular filtration rate.**
(ZIP)

## Acknowledgments

We extend our sincere gratitude to Arsi University College of Health Sciences and Asella Referral and Teaching Hospital for their invaluable support in facilitating access to the data essential for this research. We are especially grateful to all the study participants for their time and cooperation, and to the dedicated data collectors and supervisors whose efforts were instrumental to the success of this work.

## Author contributions

**Conceptualization:** Tamiru Adugna Dadi, Teshome Tola Bedada, Legesse Tadesse Wodajo, Sebona Girma Moges, Wubshet Abraham Alemu.

**Data curation:** Tamiru Adugna Dadi, Teshome Tola Bedada, Abdurke Dido Akako, Legesse Tadesse Wodajo, Sebona Girma Moges, Wubshet Abraham Alemu.

**Formal analysis:** Tamiru Adugna Dadi, Teshome Tola Bedada, Legesse Tadesse Wodajo, Sebona Girma Moges, Wubshet Abraham Alemu.

**Investigation:** Tamiru Adugna Dadi, Teshome Tola Bedada, Abdurke Dido Akako, Legesse Tadesse Wodajo, Wubshet Abraham Alemu.

**Methodology:** Tamiru Adugna Dadi, Teshome Tola Bedada, Abdurke Dido Akako, Legesse Tadesse Wodajo, Sebona Girma Moges, Wubshet Abraham Alemu.

**Resources:** Tamiru Adugna Dadi, Teshome Tola Bedada, Abdurke Dido Akako, Legesse Tadesse Wodajo, Sebona Girma Moges, Wubshet Abraham Alemu.

**Supervision:** Tamiru Adugna Dadi, Teshome Tola Bedada, Abdurke Dido Akako, Legesse Tadesse Wodajo.

**Validation:** Tamiru Adugna Dadi, Teshome Tola Bedada, Abdurke Dido Akako, Legesse Tadesse Wodajo.

**Visualization:** Tamiru Adugna Dadi, Teshome Tola Bedada, Abdurke Dido Akako, Legesse Tadesse Wodajo, Wubshet Abraham Alemu.

**Writing – original draft:** Tamiru Adugna Dadi, Teshome Tola Bedada, Abdurke Dido Akako, Legesse Tadesse Wodajo, Sebona Girma Moges, Wubshet Abraham Alemu.

**Writing – review & editing:** Tamiru Adugna Dadi, Abdurke Dido Akako.

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
