## [Decision Letter · Decision Letter 0]

15 Aug 2025

Dear Dr. Dadi,

Thank you for submitting your manuscript to PLOS ONE. After careful consideration, we feel that it has merit but does not fully meet PLOS ONE’s publication criteria as it currently stands. Therefore, we invite you to submit a revised version of the manuscript that addresses the points raised during the review process.

We look forward to receiving your revised manuscript.

Kind regards,

Donovan Anthony McGrowder, PhD., MA., MSc

Academic Editor

PLOS ONE

Journal Requirements:

2. Please update your submission to use the PLOS LaTeX template. The template and more information on our requirements for LaTeX submissions can be found at http://journals.plos.org/plosone/s/latex .

3. Please include a new copy of Table 1, 5 and 6 in your manuscript; the current table is difficult to read. Please follow the link for more information: https://blogs.plos.org/plos/2019/06/looking-good-tips-for-creating-your-plos-figures-graphics/

**Additional Editor Comments:**

Dear Dr. Dadi, 

 Your manuscript “Impaired Estimated Glomerular Filtration Rate and Associated Factors among Adult Patients Living with HIV at Asella Referral and Teaching Hospital, Ethiopia: A Cross-Sectional Study” has been assessed by two reviewers. They have raised a number of points which we believe would improve the manuscript and may allow a revised version to be published in PLOS ONE. Their reports, together with any other comments, are below.

 If you are able to fully address these points, we would encourage you to submit a revised manuscript to PLOS ONE.

Best regards,

Prof. Donovan McGrowder

Reviewers' comments:

Reviewer's Responses to Questions

**Comments to the Author**

1. Is the manuscript technically sound, and do the data support the conclusions?

Reviewer #1: Partly

Reviewer #2: Yes

2. Has the statistical analysis been performed appropriately and rigorously?

Reviewer #1: No

Reviewer #2: Yes

3. Have the authors made all data underlying the findings in their manuscript fully available?

Reviewer #1: Yes

Reviewer #2: Yes

4. Is the manuscript presented in an intelligible fashion and written in standard English?

Reviewer #1: No

Reviewer #2: Yes

Reviewer #1: In the Introduction, the authors state that “People living with HIV (PLHIV) demonstrate diverse renal pathology, including acute kidney injury (AKI), HIV-associated nephropathy (HIVAN), chronic kidney disease (CKD), and antiretroviral therapy (ART)-induced nephrotoxicity.” Therefore, what would likely interest the readers of this paper, and what holds clinical significance, is an elucidation of the distinct characteristics associated with each underlying cause of renal impairment. This perspective, however, is lacking in the current manuscript, which I consider to be its primary limitation.

Below are some technical suggestions aimed at improving the manuscript for scientific publication:

At present, the manuscript contains a substantial number of Tables, but the data could be consolidated into a single comprehensive Table for greater clarity. For instance, in Table 6, the authors perform a univariate analysis comparing two groups: those with impaired eGFR (N=47) and those with normal eGFR (N=205). By adding a column on the left to show the Overall (N=252) data, the contents of Tables 1 through 5 could be integrated, resulting in a more streamlined presentation.

Moreover, for variables that are mutually exclusive and collectively exhaustive—such as Age >40 vs. Age <40, or Hypertension Yes/No—there is no need to list both categories in separate rows. Presenting only one category suffices and will simplify the table while maintaining clarity. This recommendation also applies to the Tables presenting multivariate analysis results.

Furthermore, the Figures in the current manuscript provide minimal informational value. For example, pie charts depicting the overall distribution of BMI or renal function are unnecessary. Figures should be used more effectively to visually aid comparisons between groups or to highlight relationships between two parameters that are not easily interpreted from Tables. Enhancing the Figures in this way would greatly improve the manuscript's readability and impact.

Reviewer #2: 1.Originality

This study offers an original contribution within the context of the region in which it was conducted—Ethiopia—where data on renal function impairment among HIV-positive patients are limited. The use of the CKD-EPI 2021 formula to estimate eGFR and the detailed analysis of associated risk factors represent both methodological and scientific strengths. This approach provides an updated perspective on a clinically relevant issue in sub-Saharan Africa.

2. Importance

HIV prevalence is high in sub-Saharan Africa, and the risk of progression to chronic kidney disease (CKD) among HIV-positive patients is significant. In this context, the study is particularly important as it highlights the need for renal screening and preventive interventions in a vulnerable population. The findings may contribute to improving care protocols and shaping locally adapted public health policies.

3. Materials and Methods

The study included 252 HIV-positive patients, whose renal function was assessed using the CKD-EPI 2021 formula. Data were collected through structured questionnaires and medical record reviews. The cross-sectional design is appropriate for estimating prevalence, and the systematic sampling of participants adds methodological robustness.

4. Results

The central finding is the prevalence of impaired eGFR (18.7%), along with the identification of significant risk factors: age over 40 years, smoking, opportunistic infections, diabetes mellitus, and hypertension. These associations support the need for targeted risk factor correction and further evaluation of HIV-positive patients to prevent progression of kidney disease.

5. Additional Questions

a.What is the average duration of antiretroviral therapy among participants, and how does it influence renal function?

b.Are there differences between the treatment regimens used?

c.Based on these results, could a follow-up evaluation algorithm be developed?

**Do you want your identity to be public for this peer review?** For information about this choice, including consent withdrawal, please see our Privacy Policy

Reviewer #1: No

Reviewer #2: No

---

## [Author Response · Author response to Decision Letter 1]

4 Oct 2025

Point-by-Point Responses to Reviewers' Comments

Reviewer #1

Comment 1: "People living with HIV (PLHIV) demonstrate diverse renal pathology, including acute kidney injury (AKI), HIV-associated nephropathy (HIVAN), chronic kidney disease (CKD), and antiretroviral therapy (ART)-induced nephrotoxicity. Therefore, what would likely interest the readers of this paper, and what holds clinical significance, is an elucidation of the distinct characteristics associated with each underlying cause of renal impairment. This perspective, however, is lacking in the current manuscript, which I consider to be its primary limitation."

Response: We thank the reviewer for this insightful and clinically relevant observation. We fully agree that distinguishing between the various etiologies of renal impairment is crucial for both clinical management and scientific understanding. However, due to the cross-sectional design of our study, which relied on a single serum creatinine measurement, we were unable to definitively differentiate between specific renal pathologies such as AKI, CKD, or HIVAN, as these require longitudinal data or renal biopsy for confirmation. We have explicitly acknowledged this limitation in the revised manuscript.

Location in the manuscript: Page 21–22, Lines 329–335

Comment 2: Technical suggestions regarding tables.

Response: We appreciate the reviewer’s technical suggestions. In response, we have merged Tables 5 and 6 into a single consolidated table to improve clarity. The structure of the remaining tables has been retained to ensure they remain comprehensive yet accessible to a broad readership.

Location in the manuscript: Page 17–18, Lines 264–267

Comment 3: Technical suggestion regarding figures.

Response: We thank the reviewer for the valuable feedback on the figures. Following re-evaluation, we have removed Figure 1 to streamline the presentation. The remaining figures have been retained to support the narrative and enhance visual clarity.

Reviewer #2

We express our sincere appreciation to the reviewer for the time and constructive comments. Below are our responses to the specific questions raised:

Question 1: What is the average duration of antiretroviral therapy among participants, and how does it influence renal function?

Response: The present study did not assess the average duration of ART exposure or its potential impact on renal function. This limitation has been explicitly stated in the revised manuscript to provide clarity for readers.

Location in the manuscript: Page 21, Lines 324-327.

Question 2: Are there differences between the treatment regimens used?

Response: In this cohort, over 88% of participants were on first-line ART, with the majority (71.4%) receiving a TDF-based regimen. No statistically significant association was found between the type of ART regimen and impaired eGFR. This point has been clarified in the revised manuscript.

Location in the manuscript: Page 21, Lines 325–328

Question 3: Based on these results, could a follow-up evaluation algorithm be developed?

Response: As this is a single-center, institution-based study, the findings are not sufficient to establish a follow-up evaluation algorithm. We recommend further multicenter studies to develop and validate such an algorithm.

We are confident that the revisions have strengthened the manuscript and enhanced its clarity and impact. We believe the revised version now meets the high standards of PLOS ONE and hope it is found suitable for publication.

Sincerely,

Tamiru Adugna Dadi, MD

Corresponding Author

Assistant Professor of Internal Medicine

Arsi University, College of Health Sciences

---

## [Editor Report · Decision Letter 1]

15 Dec 2025

Impaired Estimated Glomerular Filtration Rate and Associated Factors among Adult Patients Living with HIV at Asella Referral and Teaching Hospital, Ethiopia: A Cross-Sectional Study

PONE-D-25-40706R1

Dear Dr. Dadi,

We’re pleased to inform you that your manuscript has been judged scientifically suitable for publication and will be formally accepted for publication once it meets all outstanding technical requirements.

Kind regards,

Donovan Anthony McGrowder, PhD., MA., MSc

Academic Editor

PLOS One

Additional Editor Comments (optional):

Dear Dr. Dadi,

 The manuscript entitled “Impaired Estimated Glomerular Filtration Rate and Associated Factors among Adult Patients Living with HIV at Asella Referral and Teaching Hospital, Ethiopia: A Cross-Sectional” was revised in accordance with the reviewers’ comments and is provisionally accepted pending final checks for formatting and technical requirements.

Regards,

Prof. Donovan McGrowder (Academic Editor)

---

## [Editor Report · Acceptance letter]

PONE-D-25-40706R1

PLOS One

Dear Dr. Dadi,

I'm pleased to inform you that your manuscript has been deemed suitable for publication in PLOS One. Congratulations! Your manuscript is now being handed over to our production team.

Kind regards,

on behalf of

Dr. Donovan Anthony McGrowder

Academic Editor

PLOS One